# The Efficacy of Biological Control for the Suppression of the Pea Aphid (Acyrthosiphon Pisum): Does the Resistance of Alfalfa Cultivars Matter?

**DOI:** 10.3390/insects14010028

**Published:** 2022-12-27

**Authors:** Xiang Zhang, Qiong Wu, Jianing Mu, Zunqi Chao, Qi He, Ting Gao, Chuan Wang, Mark R. McNeill, Zhaozhi Lu

**Affiliations:** 1The First-Class Discipline of Prataculture Science of Ningxia University (No. NXYLXK2017A01), College of Agriculture, Ningxia University, Yinchuan 750021, China; 2Protection Institute, Ningxia Academy of Agriculture and Forestry Sciences, Yinchuan 750002, China; 3AgResearch Ltd., Lincoln Research Centre, Christchurch 8140, New Zealand; 4Shandong Engineering Research Center for Environment-Friendly Agricultural Pest Management, College of Plant Health and Medicine, Qingdao Agricultural University, Qingdao 266109, China

**Keywords:** alfalfa, biological control, pea aphid, conservation biological control

## Abstract

**Simple Summary:**

Maximizing the combination of biological control by natural enemies and plant resistance is a key strategy for controlling aphids in various crop systems. In our study, some alfalfa cultivars were found to have resistance to the pea aphid, *Acyrthosiphon pisum*, a dominant pest in alfalfa fields in northwestern China. In addition, natural enemies were able to significantly suppress this aphid, regardless of the resistance level of the nine cultivars tested. Moreover, the biological control efficacies of predators, parasitoids, and mixed populations of natural enemies were 85%, 42%, and 88%, respectively. Conservation of natural enemies in the field is an ecologically sound and efficient approach to the management of the pea aphid in alfalfa crops.

**Abstract:**

The pea aphid, *Acyrthosiphon pisum* Harris, is a major pest of alfalfa in northwestern China. However, the roles of different groups of natural enemies in combination with aphid-resistant cultivars in the suppression of the pea aphid have not been clarified under field conditions. In this study, we used experimental cages to better understand the top-down (natural enemies) and bottom-up (nine alfalfa cultivars) biological processes, as well as the individual roles of the two processes, in the control of the pea aphid. There was a significant difference in resistance among cultivar classes revealed when natural enemies were excluded. The functional contribution of top-down suppression was higher than the bottom-up process, with natural enemies significantly suppressing aphid populations, regardless of the resistance level of different alfalfa cultivars. The mean biological efficacies of predators, parasitoids, and mixed populations of natural enemies were 85%, 42%, and 88%, respectively. Overall, our study indicated that natural enemies play a critical role in suppressing aphid populations, especially in the summer, whereas cultivar resistance did not combine effectively with natural enemies to inhibit the growth of aphids. Conservation biological control (CBC) can be implemented in the alfalfa production regions in northwestern China to reduce the overreliance on insecticides for the control of pests and mitigate their harmful effects on humans, ecosystems, and biodiversity.

## 1. Introduction

Integrated pest management (IPM) is a rational approach to pest control that protects economic, environmental, and human health interests [1]. Two distinct strategies have been promoted in IPM programs: natural enemies and entomopathogens provide top-down suppression of pest herbivore populations [2] and the alternative approach (bottom-up) involves the addition of external inputs (irrigation, fertilizer, and plant-resistant cultivars) [2,3] to minimize the impacts of insect pests and maintain production and the quality of forage dry matter. In most cases, a combination of both the top-down and bottom-up approaches is a superior approach for suppressing the abundance of pests and the damage they cause [4]. Conservation biological control (CBC) aims to improve pest regulation through the conservation of natural enemies in agroecosystems [5] and is valued at US $619/ha globally across multiple biomes, with a study conservatively estimating that CBC of endemic USA crop pests was worth $4.49 billion/year [6,7].

As an essential forage legume, alfalfa (*Medicago sativa* L.) is not only an indispensable component of the diet of livestock but also a preferred plant for ecological restoration and soil improvement in arid regions [8,9]. The Ningxia Region is located in the farming-pastoral zone in northwestern China, and it confronts two issues: the shortage of forage to feed domestic livestock and the need for ecological restoration. The livestock industry is one of the main local industries, with the area of alfalfa being grown being approximately 0.4 million hectares, accounting for more than 66% of the total area of forage [10,11]. Alfalfa planting can satisfy the requirements for forage and ecological restoration locally, but aphids, which are one of the most devastating pests of alfalfa, are estimated to cause production losses of 30% to 40% p.a. in the region [12], with the pea aphid, *Acyrthosiphon pisum* (Harris) (Hemiptera: Aphididae), being the dominant species. As a global legume pest, pea aphid has a broad host range, a complex life cycle, including both sexual and parthenogenetic reproduction [13], and overlapping generations [14], and its flexibility in adapting to different environmental conditions [15] makes it difficult to control. Traditionally, producers have relied heavily on insecticides to suppress aphids [16,17] but this practice has side effects, such as insecticide resistance [18], the loss of biodiversity, including the loss of natural enemies [16], and increased risk to human health [19]. 

There has been no systematic study in the Ningxia Region on the role and impact of the ‘bottom-up’ (cultivars) and specific ‘top-down’ (predators and parasitoids) approaches to help inform strategies to manage aphids in the alfalfa cropping system. Therefore, in the present study, by using various types of natural enemy exclusion cages, we aimed to (i) evaluate the resistances of the nine main alfalfa cultivars in the region, (ii) evaluate the role of biological control efficacy (BCE) of natural enemies (predators and parasitoids), and (iii) identify whether there is an interaction between cultivar classes and natural enemies in the suppression of pea aphid populations. The outcome of this study will help optimize the IPM of pea aphid in alfalfa production systems.

## 2. Materials and Methods

### 2.1. Experiment Design and Layout in the Field

Field experiments were conducted at the alfalfa breeding station of Ningxia Academy of Agriculture and Forestry Science in Litong District, Wuzhong City, Ningxia, China (106.1° E, 37.8° N). This area is mainly silt soil and irrigated with Yellow River water, which is a standard agricultural practice in the Ningxia agricultural area. There are nine main alfalfa cultivars grown in the Ningxia Region (Table 1). We established three replicate plots of each cultivar in May 2019. Each plot was 10 by 10 m in area and the 27 plots were randomly arranged. During the study period (June and July 2019), the average daily temperature and humidity were 23 °C and 50%, respectively (https://en.tutiempo.net/yinchuan.html, accessed on 20 September 2021). No insecticides were applied to the study sites. The management practices on the research site were based on the local schedule for growing alfalfa. 

In the Ningxia Region, there are usually four harvest times around 20 May, 30 June, 10 August, and 10 September, respectively. The current study was carried out during the second (June) and third (July) harvests. Earlier surveys [12,20] and our preliminary experimental investigation showed that natural enemies are divided into two groups of predators and parasitoids, but did not involve any investigation of entomopathogens. Predators include Coccinellids (*Hippodamia variegate* Goeze, *Propylaea japonica* Thunberg, *Harmonia axyridis* Pallas, and *Coccinella septempunctata* Linnaeus), *Syrphidae, Araneae,* Chrysopa (*Chrysoperla nipponensis* Okamoto and *Chrysopa phytochrome* Wesmael), *Anthocoridae*, and parasitoids, mainly Braconidae. The aphid population comprised a mixed community that included *Aphis craccivora* Koch (cowpea aphid), *Therioaphis trifolii* Buckton (spotted alfalfa aphid), and *A. pisum* (pea aphid) with the latter aphid the dominant species, representing more than 80% of the aphid population in the study area. Therefore, we used representative dominant species *A. pisum* in the experiments.

Two different types of cages and an open system were employed to measure the effects of natural enemies on *A. pisum*. The cages were constructed of stainless-steel tube frames (0.5 × 0.5 × 1 m, length × width × height) covered with a white, nylon net with mesh dimensions that varied according to the various experimental treatments. These were: (1) Exclusion cages covered in a fine nylon mesh with 0.53 × 0.53 mm openings which provided aphids with full protection from all-natural insect enemies. (2) Restricted access cages covered in nylon mesh with 3 × 3 mm openings in which aphids were partially protected as the size of the opening excluded the entry of large predators, e.g., coccinellids and syrphids, but allowed tiny parasitoids to enter. One side of each mesh cover for the exclusion and restriction cages was equipped with a zipper to enable entry for the addition of aphids and sampling. (3) No cage was a completely open area, with four wooden sticks standing upright in the ground to guide position, plot size, and sampling range (hereafter referred to as ‘open field’) (Figure 1a). This treatment effectively served as a positive control for the cage treatments. Within these cages or the open field, there were approximately three plants.

The three treatments were established on 10 June and again on 15 July, using a completely randomized design within each alfalfa cultivar (Figure 1b). The distance between treatments inside each plot was 3 m and between blocks was 10 m. There were three replicates for each cultivar. 

Previous studies [21] have shown that the presence of cages did not affect aphids on day 7, but began to affect aphids on day 14. To minimize the effect of cages on the aphid population and dispersal, we limited the experiment time to 14 days. Before the field experiments were established, aphids and other insects were removed from the cages and open-field plots by hand using artists paint brushes and pooters. On 10 June (second crop harvest) and 15 July (third crop harvest), each cage covered three alfalfa plants and 30 fourth instar or adult apterous *A. pisum* were placed on the highest central leaf of the plants using a camel hair paint brush, representing common summer infestation levels. From June 10 to June 26, 2019, and July 15 to July 31, 2019, we opened the zipper on each cage every seven days and shook the plants by hand so that the aphids fell onto a white plastic plate (450 mm × 600 mm) placed under the plants. The aphids, both apterous and adult, were counted and then returned to the cage so subsequent experiments were not affected by them.

### 2.2. Data Analysis

#### 2.2.1. Evaluation of Alfalfa Resistance

The aphid resistance/tolerance of cultivars was compared by using the peak aphid populations in the exclusion cages at the completion of the experiment (14 d) for both June and July bioassays. One-way ANOVA was used to compare aphid ratios calculated using the following Equation (1) [20,23] and peak populations among cultivars in the entire experiments.
(1)$$Aphid\;ratio=\frac{Mean\;aphids\;per\;cultivar}{Mean\;aphid\;number\;per\;plant\;of\;total\;obversed\;cultivars}\;$$

#### 2.2.2. Efficacy of Biological Control 

The inhibitory effect of mixed natural enemy (predators and parasitoids) populations on aphid populations was determined by comparing the exclusion-cage and open-field data sets, and the restriction-cage data was compared with the exclusion-cage and open-field data to determine the effects of parasitoids and predators on aphids, respectively.

Three analyses were conducted to assess the efficacy of the biological control of natural enemies against *A. pisum*.

Firstly, to quantify the influence of natural enemies on *A. pisum* populations, we calculated a biological control efficacy (BCE) value, calculated using the following Equation (2) [24]:(2)$$BCE=\left(1-\frac{N_{t}}{N_{0}}\right)\times 100\%$$
where *N_t_* is the number of aphids in the treatment, and *N*_0_ is the number of aphids on each cultivar without natural enemies present. Biological control efficacy values were calculated from aphid data 7 and 14 days after the establishment of the treatment.

Secondly, we assumed that the growth rate of aphid populations was exponential within the limited window of the duration of the experiment. Population growth rate (PGR) was calculated using the following Equation (3) [25]:(3)$$PGR=\frac{\ln N_{t}-\ln N_{0}}{\Delta_{t}}$$

*N_t_* is the number of aphids at the end of the experiment, *N*_0_ is the number of aphids at the beginning of the experiment, ∆*_t_* is the duration of the experiment (7th or 14th days), and ln is the natural logarithm.

Thirdly, we used insect-days as an index of the efficacy of the natural enemies. Cumulative insect-days were calculated by sequentially summing the individual insect-days. The insect-days Formula (4) [26] for the area under the curve was:(4)$$Insect-day=\left(X_{i+1}-X_{i}\right)\left(\frac{Y_{i+1}+Y_{i}}{2}\right)$$
where *X_i_* and *X*_*i*+1_ are adjacent points in time, and *Y_i_* and *Y*_*i*+1_ are the corresponding numbers of insects at those points in time. The percentage of cumulative insect-days between the treated and untreated groups indicates the percentage reduction in the number of insects.

A general linear model (GLM) was used to distinguish between the impact of alfalfa cultivars and natural enemies on aphid population growth, biological control efficacy, and the percentage reduction in cumulative insect-days (alfalfa cultivar classes and natural enemies were fixed factors). The difference between the natural enemy treatments under the same alfalfa conditions was compared with the Tukey honestly significant difference (HSD) test. Significant differences among treatments were set at the *p* < 0.05 level. Origin 2021 (Origin 2021, OriginLab Corporation, Northampton, MA, USA) was used to plot the experimental results.

## 3. Results

### 3.1. Evaluation of Alfalfa Cultivar Resistance

There were significant differences in peak aphid populations among alfalfa cultivars (*F* = 2.754, *df* = 8, *p* < 0.05) (Table 2), whereby when natural enemies were excluded, cultivar had a significant effect on the size of the aphid population. 

Based on the level of their resistance to aphids, the nine alfalfa cultivars were categorized into three classes (Table 2). Platon, Surprising, and Algonquin were in the highly susceptible group (aphid ratio > 1.25), which was significantly different from the other six cultivars, Gannong No.3, Zhongmu No.3, and Gannong No.4, which were deemed to be susceptible (aphid ratio between 0.76 and 1.25), and Golden Queen, Santory, and Crown were deemed medium resistance cultivars (aphid ratio between 0.51 and 0.75).

### 3.2. Aphid Population Dynamics in the Field

From June 10 to 25, 2019, the aphid populations in the three treatments showed a rapid increase in numbers. From July 15 to 31, 2019, only the restricted access cage showed continuous population growth, while both the exclusion cage of medium resistance and open field showed a general increase in numbers to a peak, followed by a decrease in numbers (Figure 2).

The mean number of aphids in the exclusion and restricted-access cages of all treatments was higher than that in the open field. In June, after 14 days on the plants, the mean aphid populations on three cultivar classes in exclusion cages, restricted-access cages, and open field increased from 447 to 570 aphids/plot, 302 to 455 aphids/plot, and 50 to 80 aphids/plot, respectively. In July, the number of aphids peaked from 568 to 659 aphids/plot, 348 to 487 aphids/plot, and 73 to 115 aphids/plot, respectively, 14 days after aphids were introduced (Figure 2).

### 3.3. Evaluation of Biological Control Efficacy in the Field

#### 3.3.1. Effects of Cultivar Classes and Cages on *A. pisum* Population Growth Rate (PGR)

In June 2019, there were significant differences in PGR among cultivar classes and cages (one week after June 10: Cultivar classes, *F* = 15.354, *df* = 2, *p* < 0.05; Cages, *F* = 131.219, *df* = 2, *p* < 0.05, and two weeks after June 10: Cultivar classes, *F* = 7.800, *df* = 2, *p* < 0.05; Cages, F = 305.741, *df* = 2, *p* < 0.05) (Table 3). The interaction of the cages and cultivar classes on aphid population growth rate was not significant (one week after June 10: *F* = 0.527, *df* = 4, *p* > 0.05, and two weeks after June 10: *F* = 0.506, *df* = 4, *p* > 0.05) (Table 3).

In July 2019, the PGR of aphids in the cage was significantly different among cages and cultivar classes (one week after July 15: Cultivar classes, *F* = 7.221, *df* = 2, *p* < 0.05; Cages, *F* = 170.708, *df* = 2, *p* < 0.05, and two weeks after July 15: Cultivar classes, *F* = 7.771, *df* = 2, *p* < 0.05; Cages, *F* = 316.952, *df* = 8, *p* < 0.05) (Table 3). The interaction of the cage and cultivar classes on aphid population growth rate was not significantly different (one week after July 15: *F* = 0.708, *df* = 4, *p* > 0.05, and two weeks after July 15: *F* = 1.030, *df* = 4, *p* > 0.05) (Table 3).

Population growth rates within the exclusion cage, restricted access cage, and open field were 0.19 to 0.41, 0.16 to 0.36, and 0.02 to 0.19, respectively (Figure 3). The experiments in both June and July of 2019 showed a similar pattern of aphid PGR across all different cages (open field < restricted access cage < exclusion cage) and cultivar classes (highly susceptible < medium resistance < susceptible). Apart from the open field in June, the PGR of aphids was higher after seven days than at fourteen days across all treatments. Population growth rates were significantly higher in July than in June across all treatments (Table 3).

#### 3.3.2. Effects of Cultivar Classes and Natural Enemies on *A. pisum* Biological Control Efficacy

In both June and July, the BCE was significantly different among natural enemies (one week after June 10: *F* = 83.197, *df* = 2, *p* < 0.05, two weeks after June 10: *F* = 113.379, *df* = 2, *p* < 0.05, one week after July 15: *F* = 25.211, *df* = 2, *p* < 0.05 and two weeks after July 15: *F* = 192.364, *df* = 2, *p* < 0.05) (Table 3). Cultivar classes did not significantly differently affect aphid populations under all experimental circumstances, except for the two weeks after July 15 (*F* = 3.567, *df* = 2, *p* < 0.05) (Table 3).

The BCEs of mixed natural enemy populations, predatory natural enemies, and parasitoids were 68% to 88%, 59% to 85%, and 11% to 42%, respectively (Figure 4). In the experiments in June and July of 2019, the BCEs for various combinations of natural enemies showed the same tendency: parasitoids < predators < mixed enemies. Also, the efficacy of predators and mixed enemy populations was higher two weeks after the start of the experiments than after one week, while parasitoids showed this tendency only in June on the medium resistance and susceptible cultivars. In addition, except for predators and mixed natural enemies after two weeks, the biological control efficacy of all treatments was higher in July than in June. 

#### 3.3.3. Effects of the Percentage Decrease of Cumulative Insect-Days (CIDs) for Cultivar Classes and Natural Enemies on *A. pisum*

In both experiments, there were significant differences in the percentage decrease in CIDs among top-down suppressors (predators and parasitoids) (one week after June 10: *F* = 67.383, *df* = 2, *p* < 0.05, at two weeks after June 10: *F* = 158.115, *df* = 2, *p* < 0.05, at two weeks after July 15: *F* = 24.424, *df* = 2, *p* < 0.05, and at two weeks after July 15: *F* = 367.170, *df* = 2, *p* < 0.05) (Table 3). There were no significant differences except for the observation two weeks after July 15 (*F* = 7.505, *df* = 2, *p* < 0.05) (Table 3). In addition, the interactions among the top-down (predators and parasitoids) and bottom-up (cultivar) suppressors were not significant, except for the observation at two weeks after July 15 (*F* = 2.964, *df* = 4, *p* < 0.05) (Table 3).

The percentage reductions in the CIDs of mixed populations of natural enemies, predators, and parasitoids were 57% to 88%, 51% to 84%, and 9% to 48%, respectively, across all treatments (Figure 5). Overall, the percentage decrease of CIDs was higher after two weeks than after one week in June and July, except for the parasitoid/restricted access treatments.

## 4. Discussion

This study showed that natural enemies significantly decreased aphid populations in alfalfa. The BCE of mixed populations of natural enemies ranged from 68% to 88%. In addition, the BCE of predators and parasitoids was 85% and 42%, respectively. Furthermore, predatory natural enemies were more effective than parasitoids. In the absence of natural enemies, the effects of alfalfa cultivars on the size of aphid populations were significantly different. In contrast, the effects of alfalfa cultivars on the size of aphid populations were not significantly different when natural enemies were present.

Insect resistance of plants can be divided into direct defense and indirect defense. Among many factors that affect the evaluation of plant resistance, natural enemies are an important factor [27]. To eliminate the influence of natural enemies, we used exclusion cages in the field to evaluate the resistance level, and the results of the field resistance level of alfalfa cultivars were completely consistent with the results of the field survey of Ma [20] and He [23], but there were differences with the results of Wang [28] and Liu [29]. For example, Algonquin was highly susceptible in this paper and highly resistant in their experiments. Field identification of alfalfa resistance to aphids is affected by many factors, such as field temperature and humidity, environment, rainfall, natural enemies and survey time interval, and other factors, and, in the same alfalfa cultivars planted in different environments, there are some differences in the ability to resist aphids [27,30].

The natural enemies and resistant cultivars significantly inhibit pests [31,32]. In our experiment, results from the two samplings in June and for the one week after July 15 demonstrated that the impact of cultivars was not significant when the cultivars and natural enemies were combined (see Table 3). The reason for this result may be that in the early stage, there is a high natural enemy-to-prey ratio, which would likely cause a high level of pest suppression and obscure the effect of cultivars. 

The findings of our study did not coincide with studies [33,34,35]. In these studies, bottom-up effects played a predominant role in controlling herbivore populations in biological systems. However, in the case of alfalfa cultivation, top-down suppression of an herbivore (pea aphid) by natural enemies was significantly stronger than bottom-up (cultivar) suppression, as reported in some case studies [36,37,38]. However, in the present study, there were significant differences in the inhibition of aphid populations’ interaction between cultivars and natural enemies in the two weeks after July 15, and there were also significant differences among cultivars only in this period (Table 3). This result may be due to higher mortality in the condition of high temperature, especially on susceptible cultivars (W.Q., unpublished data), which led to a significant decrease in aphids on alfalfa cultivars during this high-temperature period (Figure 2d).

Earlier studies have identified the BCEs of key natural enemies in agroecosystems but did not identify the dominant groups of key natural enemies during the different growth periods of crops [39,40]. In our experiments, predators were more effective over a longer period in both months. However, the control efficiency of the parasitoids was better in the later stage of crop growth. Predators can feed intensively on aphids later in the growing season when conditions allow for the rapid expansion of aphid populations, significantly reducing the size of aphid populations and therefore enhancing the efficiency of their control [41,42,43,44,45]. The poor control of pests by parasitoids observed at an early growth stage in alfalfa fields could be due to the unsuccessful development of parasitoid wasps in aphids, the presence of hyper-parasitoids or interspecific interactions in aphid-natural enemy communities that are detrimental to aphid parasitism by wasps [41,46]. 

In this study, three approaches were used to assess the effectiveness of the biological control of aphids by a combination of natural enemies and cultivars, including population growth rate (Figure 3), the biological control efficacy (Figure 4), and the reduction of cumulative insect-days (Figure 5). The first two methods are traditional measures of natural enemies’ suppression of aphid populations and less often considered is the function of time in shaping insect pest populations. However, the use of insect-day as a unit may provide an appropriate solution for this problem by considering the time function and the population size simultaneously [26,47,48]. We suggest all these approaches can be combined for the assessment of the management efficacy of biological control, insecticides, and natural stressors. 

Overall, our study confirmed the fundamental importance of natural enemies in the suppression of the pea aphid, *A. pisum*, in alfalfa fields in the Ningxia Province of China. Moreover, the economic threshold should be considered when making decisions for the rational spraying of chemical pesticides so that existing biological control functions are minimally harmed. In a follow-up study, it would be appropriate to consider the effects of landscape composition on the diversity of natural enemy populations and the biological control efficacy associated with the field.

## Figures and Tables

**Figure 1 insects-14-00028-f001:**
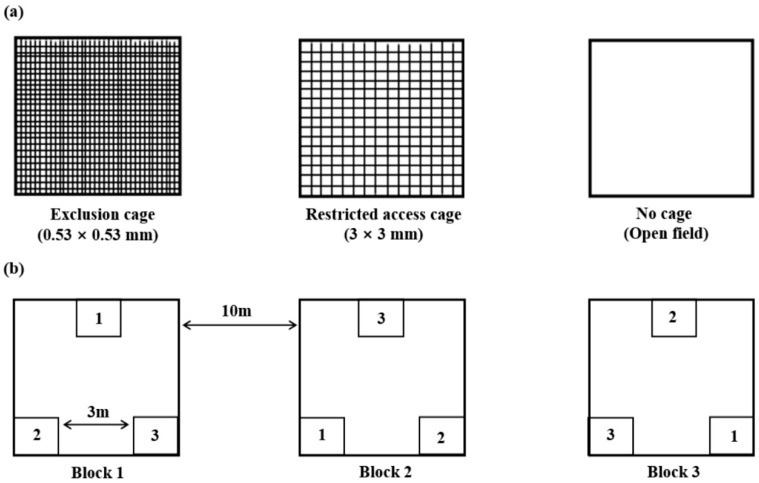
The device (**a**) and layout (**b**) of treatments in a field study in Ningxia Province, China, designed to assess the effects of the exclusion, restricted access, or full access of predators and parasitoids to the pea aphid, *Acyrthosiphon pisum* (cage design adapted from Han and Desneux 2014 [22]).

**Figure 2 insects-14-00028-f002:**
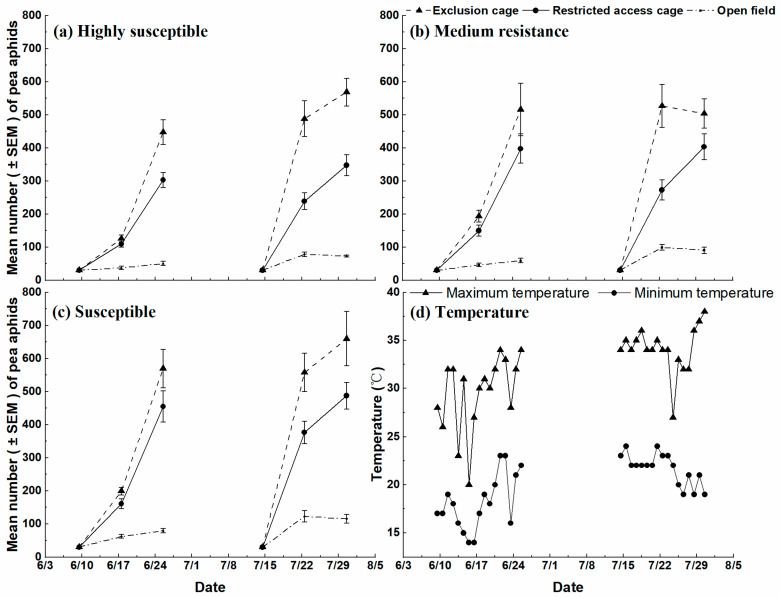
Population dynamics of pea aphid, *Acyrthosiphon pisum*, showing the mean aphid number (±SEM) on nine different alfalfa cultivars and assigned to three cultivar classes (**a**) highly susceptible, (**b**) medium resistance and (**c**) susceptible) under field conditions in Ningxia Province, China, in 2019. (**d**) The daily maximum (triangle) and minimum (circle) temperatures recorded during the experiment.

**Figure 3 insects-14-00028-f003:**
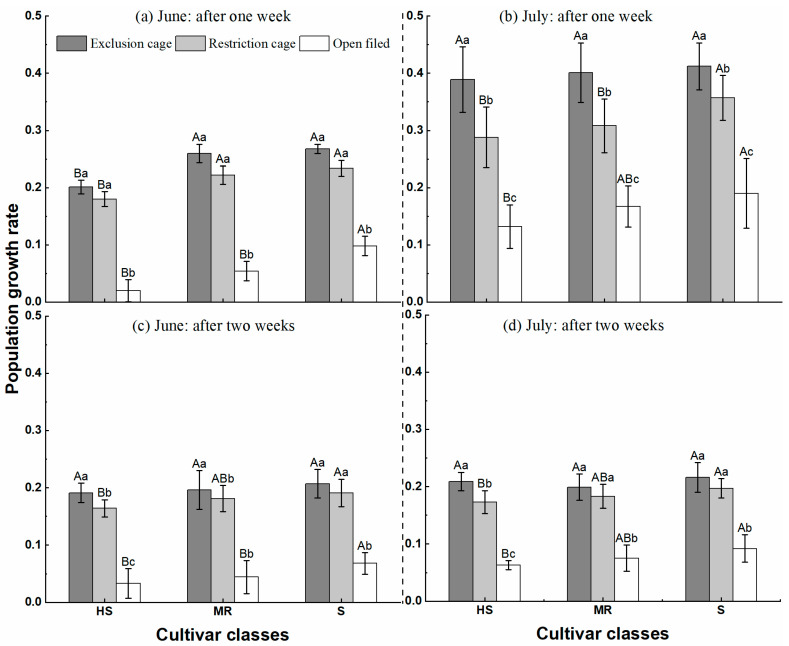
The mean (±SEM) population growth rate of pea aphid, *Acyrthosiphon pisum*, across nine different alfalfa cultivars and three caging treatments and categorized by three cultivar classes (HS = Highly susceptible; MR = Moderate Resistance; S = Susceptible) to aphids in a field experiment in Ningxia Province, China, in June (1st harvest) and July (2nd harvest) 2019. **Note:** The left side of the dashed line represents the period in June after the first harvests, and the right side represents the period in July after the second harvest. Different capital letters indicate significant differences among cultivar classes (*p* < 0.05), and different lower-case letters show significant differences among cage (*p* < 0.05).

**Figure 4 insects-14-00028-f004:**
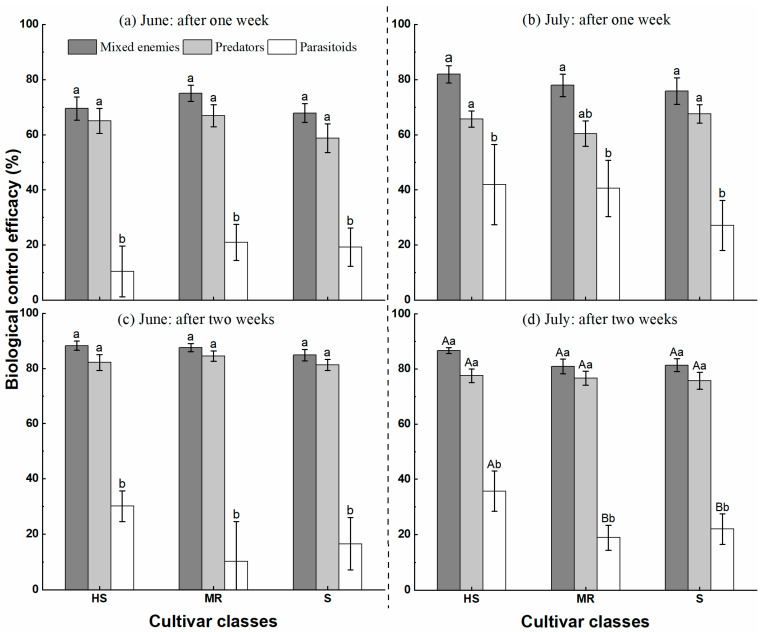
The mean (±SEM) biological control efficacy of natural enemies against the pea aphid, *Acyrthosiphon pisum*, on three cultivar classes (HS = Highly susceptible; MR = Moderate Resistance; S = Susceptible) in a field experiment in Ningxia Province, China, in 2019. **Note:** The left represents the period in June after the first harvest, and the right represents the period in July after the second harvest. Different capital letters indicate significant differences among cultivar classes (*p* < 0.05), and different lower-case letters show significant differences among natural enemies (*p* < 0.05). According to Table 3, capital letters were marked only when they were significant between treatments (HS = Highly susceptible; MR = Moderate Resistance; S = Susceptible).

**Figure 5 insects-14-00028-f005:**
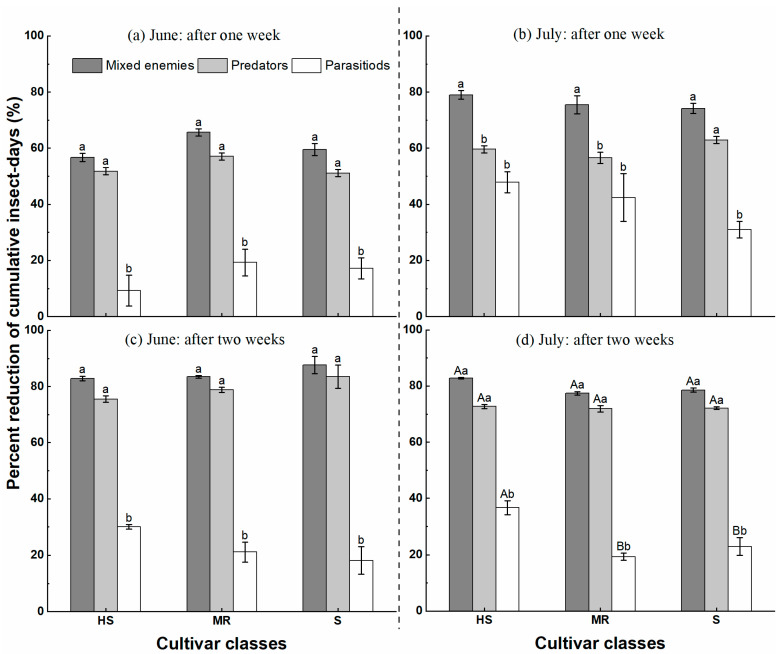
Mean (±SEM) percentage reduction of cumulative insect-days (CIDs) for natural enemies of the pea aphid, *Acyrthosiphon pisum*, on three cultivar classes (HS = Highly susceptible; MR = Moderate Resistance; S = Susceptible) in a field experiment in Ningxia Province, China, in 2019. **Note:** The left side of the dashed line represents the period in June after the first harvests, and the right side represents the period in July after the second harvest. Different capital letters indicate significant differences among cultivar classes (*p* < 0.05), and different lower-case letters show significant differences among natural enemies (*p* < 0.05), mark letters only when they were significant between treatments (HS = Highly susceptible; MR = Moderate Resistance; S = Susceptible).

**Table 1 insects-14-00028-t001:** Alfalfa were used in a biological control field study involving the pea aphid, *Acyrthosiphon pisum*, in Ningxia Region, China, in 2019.

Cultivar	Source	Year of Introduction
Platon	Germany	2011
Surprising	Canada	2011
Algonquin	Canada	2012
Gannong No.4	Grassland Science College of Gansu Agricultural University	2012
Zhongmu No.3	Beijing Institute of Animal Science of Chinese Academy of Agricultural Science	2012
Gannong No.3	Grassland Science College of Gansu Agricultural University	2012
Golden Queen	Canada	2012
Santory	France	2011
Crown	America	2011

**Table 2 insects-14-00028-t002:** Alfalfa cultivars and three population parameters of the pea aphid, *Acyrthosiphon pisum*, from a biological control field study in Ningxia Region, China, in 2019.

Cultivar	Peak Population	Aphid Ratio	Resistance Level *
**Platon**	439.67 ± 96.09 B	1.49 ± 0.0093 a	I: Highly susceptible (>1.25)
**Surprising**	870.67 ± 292.74 A	1.41 ± 0.0557 b	
**Algonquin**	525.67 ± 78.83 B	1.40 ± 0.0498 b	
**Gannong No.4**	629.67 ± 75.06 AB	1.24 ± 0.0105 c	II: Susceptible (0.76~1.25)
**Zhongmu No.3**	412.00 ± 161.75 B	0.78 ± 0.0091 d	
**Gannong No.3**	667.67 ± 79.75 AB	0.75 ± 0.0131 d	
**Golden Queen**	606.67 ± 229.05 AB	0.69 ± 0.0110 e	III: Medium resistance (0.51~0.75)
**Santory**	470.00 ± 43.59 B	0.66 ± 0.0220 e	
**Crown**	573.00 ± 19.08 B	0.60 ± 0.0110 f	

**Notes:** Different upper- and lower-case letters in a column indicate a significant difference (*p* < 0.05). * The mean resistance level for each of the nine alfalfa cultivars is based on the results of the exclusion-cage treatment.

**Table 3 insects-14-00028-t003:** F-values and significance levels of the efficacy of control by predators and parasitoids of the pea aphid, *Acyrthosiphon pisum*, on alfalfa cultivar classes in a field experiment in Ningxia Province, China, in 2019.

Variables	Experimental Period (Month/Day~Month/Day)	Cultivar Classes		Cages/Natural Enemies		Cages/Natural Enemies × Cultivar Classes	
		*F*	*p*	*F*	*p*	*F*	*p*
**PGR**	After one week (6/10~6/18)	15.354	**0.000 *****	131.219	**0.000 *****	0.527	0.716
	After two weeks (6/10~6/26)	7.800	**0.001 ****	305.741	**0.000 *****	0.506	0.731
	After one week (7/15~7/23)	7.221	**0.001 ****	170.708	**0.000 *****	0.708	0.589
	Within two weeks (7/15~7/31)	7.771	**0.001 ****	316.952	**0.000 *****	1.030	0.398
**BCE**	After one week (6/10~6/18)	1.103	0.337	83.197	**0.000 *****	0.489	0.744
	After two weeks (6/10~6-26)	0.944	0.394	113.379	**0.000 *****	0.939	0.447
	After one week (7/15~7/23)	0.555	0.577	25.211	**0.000 *****	0.551	0.699
	After two weeks (7/15~7/31)	3.567	**0.033 ***	192.364	**0.000 *****	1.157	0.337
**CIDs**	After one week (6/10~6/18)	1.882	0.181	67.383	**0.000 *****	0.189	0.941
	After two weeks (6/10~6-26)	0.154	0.859	158.115	**0.000 *****	1.306	0.305
	After one week (7/15~7/23)	0.741	0.491	24.424	**0.000 *****	0.781	0.552
	After two weeks (7/15~7/31)	7.505	**0.004 ****	367.170	**0.000 *****	2.964	**0.048 ***

**Note:** F-values and significance levels are from general linear models relating to population growth rate (PGR), biological control efficacy (BCE) and the reduction of cumulative aphid-days (CIDs) of aphids to cultivar classes, cage/natural enemies, and their interactions. Significant values (*p* < 0.05) are highlighted in bold. * *p* < 0.05, ** *p* < 0.01, *** *p* < 0.001.

## Data Availability

Data used in this study are available from the corresponding authors upon reasonable request.

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
