# Peer review of "The Efficacy of Biological Control for the Suppression of the Pea Aphid (Acyrthosiphon Pisum): Does the Resistance of Alfalfa Cultivars Matter?"

_insects, 2022, doi:10.3390/insects14010028_

Round 1

Reviewer 1 Report

This paper reports exclusion experiments at a field station in NingXia, China.  The authors use nine (9) different cultivars of alfalfa with putative resistance levels and two mesh sizes for cages: one to exclude all natural enemies, and the other to exclude larger predators.  There was an open “cage”.

Such experiments are essential to assess the relative contribution of host plant resistance and the potential of natural enemies to control aphid pests.  In this case the authors concentrate on one key species in their system, the pea aphid, Acyrthosiphon pisum.

The authors present three measures of biological control “efficiency”.  They all pretty much say the same thing and unless there is to be a comparison of the advantages of using one or the other, I suggest two can be relegated to supplementary and one presented in the paper.

The authors need to be circumspect as to their findings.  At best they have measures of the potential of natural enemies as control agents derived from small plots in an experimental farm with no insecticides used.  This is far removed from real farm production systems.

My main difficulties are with the presentation and language.  In many cases I could not follow what was done.  I will not repeat the problems here as I have annotated the manuscript extensively.  These issues need to be clarified.  I note there is a native English language author listed.  Can I suggest he be consulted more closely.

Reviewer 2 Report

The paper is interesting, because the use of the natural resistance for the control of pests in alfalfa has a great importance, this being an essential crop for forage and soil restoration.

Few observations:

- row 209 ”...field increased from 447 .... ” instead of ”...field increased to 447 .... ”;

-row 210 ”... peaked from 568 ....” instead of ”... peaked at 568 ....”;

- in figures 3, 4 and 5 is displayed SE or SD? - please mention in the title;

- see rows 344-360 there something is wrong, maybe some lines were pasted there by mistake;

There are necessary some improvements of the expression, mainly in the sections Result and Discussions.

Reviewer 3 Report

Dear Authors,

The manuscript is relatively well written improvements are needed before acceptance for publication. Please take into account the following above critical points:

The methods used to conduct this study appear proper but the experimental design should be described in details. How did you record the natural enemies of the pea aphid in the cages and open field plots?

The data received were analysed correctly with the exception of parts about the natural enemies. The terms natural enemies, predators and parasitoids have been used in general. It is not clear if mixed natural enemies include pathogens. How many taxons did you record as natural enemies?

Other comments and suggestions for improvement of the manuscript are bellow and in the attached file:

Introduction

The manuscript is about the pea aphid, which is an important pest on fabaceous crops. I suggest short information about this species to be included - distribution, host plants, live cycle, impact, and management strategies and practices. There is a recent review paper of Sandhi and Reddy (2020) (doi: 10.1093/jipm/pmaa016)

Materials and Methods

Details about preceding crop and soil type at the investigation site are needed.

Line 86-87:  Geographic coordinates should be added.

Line 99-100: The authors have observed several aphid species, and Acyrthosiphon pisum has been the dominant species. How the authors distinguish the aphid species during the study?

Natural enemies include pathogens, predators and parasitoids. Do the authors mean all groups of natural enemies?

Main comment about Material and Methods section: Reading this section it is not clear how the authors record the natural enemies.

How the grade of resistance level [medium resistance (0.51 ~ 0.75), susceptible (0.76 ~ 1.25) and highly susceptible (> 1.25)] is determined/based? Add reference.

Results

3.1 Evaluation of alfalfa cultivar resistance

Lines 190-192: Results repeat the information presented in Table 2.

3.3.1 Effects of cultivars and cages on A. pisum population growth rate (PGR)

Lines 223-224 and lines 230-231: Probably cultivar classes instead of cultivars (as df = 2).

Discussion

Line 322: By my opinion BCE at 42% is moderate instead of high.

Probably the authors can discuss the results about the effect of different pea cultivars on the pea aphid (Morgan et al., 2001https://doi.org/10.1079/BER200062; He and Zhang, 2006)

Kind regards,

Reviewer

Reviewer 4 Report

Dear Authors, 

This manuscript is certainly interesting. However, there are some imprecisions and lacking information that, if corrected would greatly improve the manuscript. 

The threemain issues I see are: 

1. The cages differ in mesh size to exclude either all insects (0.5mm openings), or exclude larger insects only (3mm opening), or no exclusion (open filed). This means that in the OPEN field situation there are BOTH parasitoids and predators that can access. Please make this clear, as in some spots (such as figure 4) you seem to say ONLY predators can acsess the 3mm cages? 

2. Can you provide a list of the natural enemies that were actually observed? They should be present in the aphid samples when measuring population densities. No need to have details but at least a list of different genus observed would be useful. 

3. The cages will most likely also cause a difference in micro-climatic conditions, such as shade, higher moisture levels, higher temperatures, reduced wind. Did you collect data on this and could it be (partially) responsible for the differences observed?   

 There are also a number of minor points that could be improved: 

- The experimental design and drawing are a bit confusing as there should be only 3 treatments (not 4) in figure 1b. Please clarify

- Do I understand it correctly that 1 cage = 1 plant? Please clarify

- You are using several different metrics for aphid population development. Please explain why and how they are used differently for the interpretation of the results.

-  Table 2 APHID RATIO. Please explain in more detail as this can be calculated at several moments of the experiment. 

- Table 2 PEAK population: the data suggest a clear variation between replicates inside each cultivar. Can you explain? 

-   When setting up the experiments you 'removed insects'. How succesfull was that (were there any signs of other aphids remaining?) and did you encounter natural enemies when doing that?   

- Was there any aphid MIGRATION during the experimental period and could this have influenced your results?

I think you should be able to provide these details from your dataset, and it would increase the quality of your manuscript.   

Round 2

Reviewer 1 Report

Much improved but I still have some minor questions and suggestions.  Is reference 30 a correct citation? I think what you call plot you also call a block.  This is confusing.
